# Effects of Relational Benefits in the Model of Customers' Benefits and Relationship Quality in Vietnam

Phuong T. Nguyen [1,*], Hieu V. Cao [2], Hiep M. Phuoc [2] and Phong T. Tran [3]

1 Department of Corporate Relations and Job Placement, Nguyen Tat Thanh University, 300A Nguyen Tat Thanh Street, Ho Chi Minh City 700000, Vietnam
2 Department of Scientific Management and External Relations, Binh Duong University, 504 Binh Duong Avenue, Thu Dau Mot Town 820000, Vietnam
3 Department of Science and Technology, Long An University of Economics and Industry, 938 National Highway 1, Tan An City 850000, Vietnam
* Correspondence: phuongnt@ntt.edu.vn; Tel.: +84-913905095

**Abstract:** With the aim of comparing the influence of economic benefits with social benefits in the model of integrating customer benefits and relationship quality in the context of university–enterprise relationship research in Ho Chi Minh City (HCMC), Vietnam. From the perspective of enterprises, a study combining qualitative and quantitative research was carried out. Data for the main study were collected from 486 enterprises using an online survey. The research model and hypotheses are tested by analyzing the structural equation model. The results of examining the influence of economic benefits and social benefits in the research model indicate that the influence of economic benefits is more significant than the influence of social benefits. This is a new finding of this study in comparison with previous studies on relational benefits. In addition, the study also pointed out that economic benefits and social benefits have a direct impact on perceived service quality.

**Keywords:** customer loyalty; perceived service quality; relational benefits; relationship quality





## 1. Introduction

Regarding the motivation for enterprises to enter into the relationship, Peterson (1995) and Sheth and Parvatiyar (1995) argue that economic benefits are the main driving force for developing business-to-business (B2B) relationships. More specifically, Peterson (1995) argues that saving money is the main motivation to engage in relational exchanges. However, the results of many recent studies do not seem to support these scholars' views.

Since the study of Gwinner et al. (1998), the issue of customer benefits when participating in a relationship in the service sector received particular attention from many scholars. In particular, the relational benefit approach and relationship quality by Hennig-Thurau et al. (2002) have been applied by many studies (for example, Palaima and Auruškevičienė 2007; Gremler et al. 2020). While Palaima and Auruškevičienė (2007) studied the B2B relationship in parcel delivery, Gremler et al. (2020) conducted a meta-analysis of 224 research papers on relational benefits in the last 20 years, including 42 studies on the relationship between businesses and enterprises (B2B). The results of these authors show that special treatment benefits (including economic benefits and customization benefits) have only a very small influence on the research model compared to social benefits and confidence benefits. This result contradicts the views of Peterson (1995) and Sheth and Parvatiyar (1995). This contradiction has prompted the writer to re-examine the effect of economic interests in a similar research model.

Discussing customer benefits, Gwinner et al. (1998) argue that besides the relationship benefits, "the customer's benefits when entering into the relationship can focus on services"; because customers want to receive benefits from services, they have a relationship with suppliers (Gwinner et al. 1998), the level of meeting this type of benefits is reflected in

perceived service quality (PSQ). Thus, when researching the benefits of customers, it is necessary to pay attention to both types of benefits mentioned above. Therefore, this study inherits and extends the approach of relational benefits and relationship quality of Hennig-Thurau et al. (2002) into an integrated model of customer benefits and B2B relationship quality in the context of research on service relationship between universities and enterprises in Ho Chi Minh City (HCMC), Vietnam.

HCMC is the economic, cultural, service, and educational center of Vietnam. According to the White Paper on Vietnamese enterprises 2021, HCMC has 254,699 operating enterprises, accounting for about 37% of the number of enterprises in the country. Business relationships in Ho Chi Minh City are full of characteristics of a transition economy in the context of Asian culture. Besides, this city currently has 63 higher education institutions, accounting for about 26% of the number of higher education institutions in the country. Particularly, excluding 15 public institutions in the fields of defense, security, staff training, and fine arts, the remaining 48 higher education institutions in Ho Chi Minh City have relationships with enterprises. Therefore, HCMC is a suitable choice to study the benefits of enterprises in their relationship with universities. This selection may help uncover some differences from previous relational benefits studies that were largely based on Western industrial culture, as Gwinner et al. (1998, p. 111) have shown that "It is also quite possible that the benefits received, or their importance, in the customer—service provider relationship may be very different when considered in other cultural contexts".

The relationship between higher education institutions and enterprises not only increases the ability to transfer technology, knowledge, and meet human resource demands but also helps to form and develop innovative start-ups (Carayol 2003; Gibb and Hannon 2006). There have been many studies on the university–enterprise relationship according to the educational research approach (e.g., Etzkowitz and Leydesdorff 2000; Carayol 2003; Gibb and Hannon 2006). According to the marketing approach, there has been a number of researchers mentioning the relationship between universities and students (Holdford and White 1997; Athiyaman 1997; McCollough and Gremler 1999; Hennig-Thurau et al. 2001) or research on service relationships between universities and cultural institutions (e.g., Segarra-Moliner et al. 2013). However, the level of interaction between educational research and marketing research on this topic is rather low. Many authors have supported the view that universities can be viewed as service providers (Dolinsky 1994; Kotler and Fox 1995; Licata and Frankwick 1996; Zammuto et al. 1996; Joseph and Joseph 1997; Athiyaman 1997; Hennig-Thurau et al. 2001; Segarra-Moliner et al. 2013). Services that universities provide to businesses include training courses, internships, research projects, licensing, patents, product and service development, innovation, and more (Dan 2013).

This study considers the perspective of enterprises on the university–enterprise relationship according to the benefit and relationship quality approach in relationship marketing theory to study the effects of relational benefits in the model of customers' benefits and relationship quality in Vietnam. From there, the study compares the level of influence between economic benefits and other main relational benefits in the research model to expand understanding of customer benefit dynamics in the B2B relationship in the service sector in a transition economy in Vietnam.

## 2. Theoretical Overview and Research Hypotheses

### 2.1. Relationship Marketing (RM)

The concept of relationship marketing (RM) was first introduced by Berry. Berry et al. (1983, p. 25) stated that "relationship marketing is a strategy to attract, maintain, and enhance relationships with customers of service organizations". The basic philosophy of RM is based on the assumption that supplier-buyer interaction strategies can build and preserve buyer loyalty (Berry 1995). For a relationship to exist, it must be perceived as mutually beneficial by the partners (Barnes 1994). Berry (1995) also argued that a marketing relationship benefits the customer as well as the company. Associated with the theme of RM is relationship quality (RQ).

The university–business relationship meets the conditions for effective practice of RM: (1) the enterprise has a continuous or periodic demand for the service provided by the university; (2) Enterprises can control the choice of universities (3) There are many universities providing alternative services and it is common for enterprises to switch from one university to another. In addition, Berry et al.'s (1983) definition of RM corresponds to the university's goal of successfully building relationships with businesses. Accordingly, attracting businesses is just an intermediary step, strengthening relationships, and turning indifferent businesses into loyal customers of the university is also RM. The concepts that the study is interested in such as perceived service quality (PSQ), relational benefits (RB), relationship quality (RQ) and customer loyalty (LOY) are all core concepts of the RM theory.

*2.2. The Quality of the University–Enterprise Relationship*

Many scholars have attempted to define RQ in order to define its dimensions and determine its premises and consequences in different contexts, resulting in several different definitions of CLMQH. Among them, Smith (1998) suggested that RQ is a higher-order structure consisting of satisfaction, trust, and commitment components. This view is supported by many researchers (e.g., Vieira et al. 2008, Walsh et al. 2010; Chu and Wang 2012; Susanta et al. 2013; Purnasari and Yuliando 2015; Gremler et al. 2020; Tajvidi et al. 2021; Nguyen et al. 2021). The author conducted qualitative research to determine the components of the quality of the university–enterprise relationship, and the results supported the view of Smith (1998). The quality of the university–enterprise relationship is also a higher-order construct including satisfaction, trust, and commitment components. In this study, the quality of the university–enterprise relationship is the intermediate variable of the research model.

2.2.1. Satisfaction (SAT)

Many researchers identify satisfaction as an important component to build RQ (e.g., Dwyer et al. 1987; Crosby et al. 1990; Morgan and Hunt 1994; Skarmeas et al. 2008). According to Oliver (1999), "satisfaction is the customer's perception of the difference between previous expectations and outcomes when receiving goods or experiencing services". Doyle (2002) suggested that "a very satisfied customer will exhibit the following characteristics: (1) loyalty, stay in the relationship longer, (2) buy more, (3) positive word of mouth, (4) pay less attention to other competing brands and other advertising and (5) the company will reduce costs compared to serving new customers". Satisfaction is an important component to build the quality of the university–enterprise relationship. Accordingly, satisfaction is the positive emotional state of enterprises when evaluating the interaction aspects are well done. Enterprise satisfaction is very important because satisfied customers will often orient long-term cooperation (Jesús and Polo-Redondo 2011), hence, customers are less likely to switch to other partners (Kotler and Gertner 2002), hope to have a long-term relationship with a supplier (Storbacka et al. 1994).

2.2.2. Trust (TRU)

In the integrated model of RB and RQ of Hennig-Thurau et al. (2002), RQ has only two components, satisfaction and commitment—in which, the trust factor is studied as confidence benefits are one of three types of RB. However, the author's qualitative research shows that trust is one of the three components of the quality of the university–enterprise relationship. Therefore, this study will not include confidence benefits in the research model.

Ganesan (1994) considered "trust to be one of the most widely tested and accepted concepts in relationship marketing". Similarly, Muafi (2015, 2016) confirmed trust as an important indicator of RQ. There are a number of different interpretations of this concept, such as Dwyer et al. (1987) suggested that "trust is when one party expects the other to fulfill their obligations and responsibilities in the relationship". Similarly, Crosby et al. (1990) argued that trust is when customers are confident that they can rely on someone they believe will serve their long-term interests. More specifically, Morgan and Hunt (1994)

suggested that trust is formed when one partner in a relationship believes that their partner is trustworthy and upright. In the university–enterprise relationship, trust expresses the confidence of enterprises when they believe that this university will meet their requirements (Anderson and Weitz 1989), enterprises will try to reduce risk by how to choose a university that is considered to have credibility and benevolence.

### 2.2.3. Commitment (COM)

Berry and Parasuraman (1991) argued that the commitment of two partners to the relationship is the foundation on which the relationship is built. Defining commitment, Morgan and Hunt (1994) argued that "relationship commitment as an exchange partner believing that an ongoing relationship with another is so important as to warrant maximum efforts at maintaining it; that is, the commitment party believes the relationship is worth working on to ensure that it endures indefinitely". Assessing the importance of commitment in business relationships, Dwyer et al. (1987) argued that "commitment is the highest level of relationship linkage in the relationship development process model through five stages: (1) awareness; (2) discovery; (3) expansion; (4) commitment; and (5) dissolution". In the university–enterprise relationship, commitment is the long-term orientation of the enterprise toward the relationship with the university, a desire to maintain a long-term and valuable relationship. A commitment is formed when both the enterprise and the universities are willing to sacrifice short-term benefits for long-term benefits. Once committed, businesses in the relationship will not be willing to switch even if another university has superior incentives compared to the university that the enterprise has committed.

### 2.3. Customer Loyalty (LOY)

In the service sector, clients who have interconnection with a supplier can derive two types of benefits including benefits from the core service and benefits from the relationship itself (Gwinner et al. 1998; Hennig-Thurau et al. 2000). Indeed, considering the nature of the service transaction, because customers (enterprises) want to receive benefits from services, they connect to suppliers (universities) (Gwinner et al. 1998), the level of response of these benefits is reflected in service quality perceived (PSQ) by customers (enterprise). Besides this, close customers can also get benefits from the relationship itself, this second type of benefit is named relational benefits (RB). This study extends Hennig-Thurau et al. (2002)'s integrated model of relationship benefits and relationship quality (2002) by adding the PSQ factor to the research model.

### 2.4. Customers' Benefits

In the service sector, clients who have interconnection with a supplier can derive two types of benefits including benefits from the core service and benefits from the relationship itself (Hennig-Thurau et al. 2000). Indeed, considering the nature of the service transaction, because customers want to receive benefits from services, they connect to suppliers (Gwinner et al. 1998), the level of response of these benefits is reflected in the customer's perception of the service quality. Besides this, close customers can also get benefits from the relationship itself, this second type of benefit is named relational benefits (RB). Initial qualitative research by Gwinner et al. (1998) determined that "there are four types of relational benefits namely social benefits, psychological benefits, economic benefits, and customization benefits". Similar to the study of Nguyen et al. (2021), this study will focus on considering economic benefits and social benefits for businesses in the service relationship with universities.

### 2.5. Relational Benefits (RB)

Initial qualitative research by Gwinner et al. (1998) determined that "there are four types of relational benefits namely social benefits (SOB), psychological benefits, economic benefits (ECB), and customization benefits". Over time, more types of relationship benefits have been added such as Identity-Related Benefits (Fournier 1998), Functional Benefits

(Reynolds and Beatty 1999; Tsimonis et al. 2020), Quality Improvement Benefits (Sweeney and Webb 2002), Respect Benefits (Chang and Chen 2007), Value-Added Benefits (Li 2011; Li et al. 2012), Collaborative Benefits (Li 2011; Li et al. 2012); Hedonic Benefits (Meyer-Waarden et al. 2013), Enjoyment Benefits (Li 2011; Tsimonis et al. 2020), Self-Enhancement Benefits (Hennig-Thurau et al. 2004; Tsimonis et al. 2020), Advice Benefits (Hennig-Thurau et al. 2004; Tsimonis et al. 2020), Status Benefits (Li 2011; Tsimonis et al. 2020), Safety Benefits (Yang et al. 2017; Lee et al. 2021), Epistemic Benefits (Lee et al. 2021). However, these additional benefits do not appear frequently, so this study only focuses on analyzing the relational benefits introduced by Gwinner et al. (1998).

Although initially Gwinner et al. (1998) introduced four types of RBs, however, in the study of the relationship between businesses and individual customers (B2C), Gwinner et al. (1998) classified the four types of RBs initially into new three types including (1) Social benefits (SOB), (2) Confidence benefits, and (3) Special treatment benefits (STB), with STB including discounts, faster service, . . . (economic benefit) or special additional service. Accordingly, economic benefits are only a component of STB in the study of B2C relationships. However, Gwinner et al. (1998) noted that "future studies should explore whether similar benefits are present in the context of B2B relationships". Thus, when studying B2B relationships, it is necessary to re-examine the classification of RB because Gwinner et al. (1998) also argued that "economic benefits that customers can receive when participating in a relationship exchange is the main driver for developing relationships between businesses" (B2B relationship), as Peterson (1995) and Sheth and Parvatiyar (1995) pointed out earlier. Therefore, instead of repeatedly applying STB to B2B relationship research (e.g., Palaima and Auruškevičienė 2007; Gremler et al. 2020), the study will focus on analyzing ECB and its effects.

2.5.1. Economic Benefits (ECB)

Gwinner et al. (1998) argued that "economic benefits (ECB) include both monetary and non-monetary benefits". From the review of studies that mentioned ECB, the author has conducted qualitative research, and the results have confirmed that the components of ECB in the relationship are consistent with the theory, including cost reduction (Williamson 1988; Heide and John 1992; Kalwani and Narayandas 1995; Sheth and Parvatiyar 1995; Gwinner et al. 1998; Li et al. 2012), time savings (Gwinner et al. 1998), recover R&D costs faster (Sheth and Parvatiyar 1995), share in technology, information and market access opportunities (Wilson 1995), gain knowledge from partners (Badaracco 1991; Wilson 1995) and receive special additional services (Gwinner et al. 1998; Gremler et al. 2020). Businesses participating in qualitative research believe that ECB is very important to them, it affects their perceptions of service quality, relationship quality, and loyalty. Therefore, ECB is selected to include in the research model.

For the interrelation of ECB and perceived service quality (PSQ), Palaima and Auruškevičienė (2007) and Chen and Hu (2013) confirmed that service quality directly affects RB. However, Isen and Baron (1991) state that "feelings shape thought and thought shapes feelings". Besides this, several organizational behavior studies have demonstrated that feelings and emotions influence several important organizational behaviors (e.g., George and Brief 1996; Podsakoff and MacKenzie 1997). This suggests that benefits may have an influence on perception and raises the question of whether customers have a more positive perception of service quality when receiving relational benefits. The author's qualitative research shows that in HCMC when receiving relational benefits, businesses feel more positively about the university's service quality. Therefore, the author expects economic benefits will have a direct influence on the perception of enterprises about the university's service quality. Besides, Nguyen et al. (2021) confirmed that economic benefits have a direct influence on relationship quality (RQ) and customer loyalty (LOY). Based on the results of previous studies and the above arguments, the author proposes the following hypotheses:

**Hypothesis 1 (H1).** *Economic benefits have a direct positive impact on PSQ.*

**Hypothesis 2 (H2).** *Economic benefits have a direct positive impact on RQ.*

**Hypothesis 3 (H3).** *Economic benefits have a direct positive impact on LOY.*

2.5.2. Social Benefits

Gwinner et al. (1998) argued that social benefits (SOBs) are related to emotions such as personal recognition, becoming a loyal customer, and friendship between customer and service supplier. In service relationships, Liljander and Strandvik (1995) stated that "SOBs exist when the customer and the service employee know each other well, communicate easily and have mutual trust". In addition, customers can also benefit from social interactions through shopping (Darden and Dorsch 1990), since "service encounters are also social encounters, repeated contact naturally occurring between individuals" (Czepiel 1990), which helps "address the basic human need to feel important" (Jackson 1993) and the SOBs that arise from social relationships go beyond ECB (Hennig-Thurau et al. 2002). Enterprises participating in qualitative research believe that a relationship with the university can bring them some SOB such as personal recognition, being treated with respect, developing friendships, to continue to access other opportunities through the relationship and many enterprises expressed their interest in social aspects of this relationship. Enterprises think that receiving SOB is important to them, it affects their perception of service quality, relationship quality, and loyalty. Therefore, SOB was selected to be included in the research model.

With regard to the influence of SOB on perceived service quality (PSQ), many researchers believe that the social relationship between service providers and customers can be a powerful tool to enhance customers' perception of the benefits of core services (e.g., Crosby 1989; Kempeners 1995; Price and Arnould 1999). From the confirmation of these authors and the argument about the influence of economic benefits on PSQ in the above section, the author believes that SOBs have a significant effect on the perception of enterprises about the university's service quality. Besides this, Nguyen et al. (2021) confirmed that SOBs have a direct influence on relationship quality (RQ) and customer loyalty (LOY). Based on the results of previous studies and the above arguments, the author proposes the following hypotheses:

**Hypothesis 4 (H4).** *Social benefits have a direct positive impact on PSQ.*

**Hypothesis 5 (H5).** *Social benefits have a direct positive impact on RQ.*

**Hypothesis 6 (H6).** *Social benefits have a direct positive impact on LOY.*

*2.6. Perceived Service Quality (PSQ)*

Grönroos (1982) argued that the quality of a particular service is viewed as the result of the service users' perceived evaluation process when comparing their expectations with the actual service they experience, the result of that process is perceived service quality (PSQ). Similarly, Parasuraman et al. (1988) argued that PSQ is the perception of customers as a result of the comparison between their expectations and the service they actually receive. On the basis of the synthesis of definitions, it can be expressed that the service quality of the university as perceived by enterprises is the result of the evaluation process according to the perception of enterprises when comparing the actual service experience provided by the university with their expectation of benefits. The author's qualitative research shows that university service quality as perceived by businesses includes three components: technical quality (what the service is provided), functional quality (how the service is provided), and the image of the university as perceived by enterprises. This result is consistent with the PSQ model of Grönroos (1993) with three components: "technical quality", "functional quality", and "image".

Discussing the influence of perceived service quality on relationship quality, Crosby et al. (1990) argued that service quality is a condition for RQ; Rauyruen and Miller (2007) even suggested that service quality is one of the components of RQ. Similarly, many studies confirm service quality as one of the factors affecting RQ (e.g., Crosby et al. 1990; Lagace et al. 1991; Wray et al. 1994; Bejou et al. 1996; Parsons 2002; Palaima and Auruškevičienė 2007). In addition, Hennig-Thurau et al. (2001, pp. 335, 337) identified that "educational service quality, according to student perception (PSQ), has a significant positive impact on student loyalty". Similarly, Palaima and Auruškevičienė (2007) also determined that "service quality has a direct influence on customer loyalty". From the results of previous studies, the author believes that the quality of university service as perceived by enterprises affects the quality of university–enterprise relationship and the loyalty of enterprises and proposes the following hypotheses:

**Hypothesis 7 (H7).** *Perceived service quality has a direct positive impact on RQ.*

**Hypothesis 8 (H8).** *Perceived service quality has a direct positive impact on LOY.*

*2.7. The Effect of University–Enterprise Relationship Quality*

As mentioned, the quality of the university–enterprise relationship includes three components: satisfaction, trust, and commitment, these three components play different roles in developing the relationship (Aurier and N'Goala 2010); in which, commitment requires prior trust, and both are motivated by satisfaction (Segarra-Moliner et al. 2013). Prince et al. (2016) argue that relationship quality is an important variable to achieve customer loyalty (LOY). Similarly, many studies have identified LOY as one of the outcomes of RQ (e.g., Liu et al. 2011; McDonnell et al. 2011; Aurier and Lanauze 2011; Li et al. 2012; Yang et al. 2017; Gremler et al. 2020). Based on the results of previous studies, the author believes that the quality of the university–enterprise relationship has a direct positive influence on the loyalty of enterprises and proposes the hypothesis:

**Hypothesis 9 (H9).** *Relationship quality has a direct positive impact on LOY.*

**3. Research Models**

The following model presents hypothetical relationships (see Figure 1).

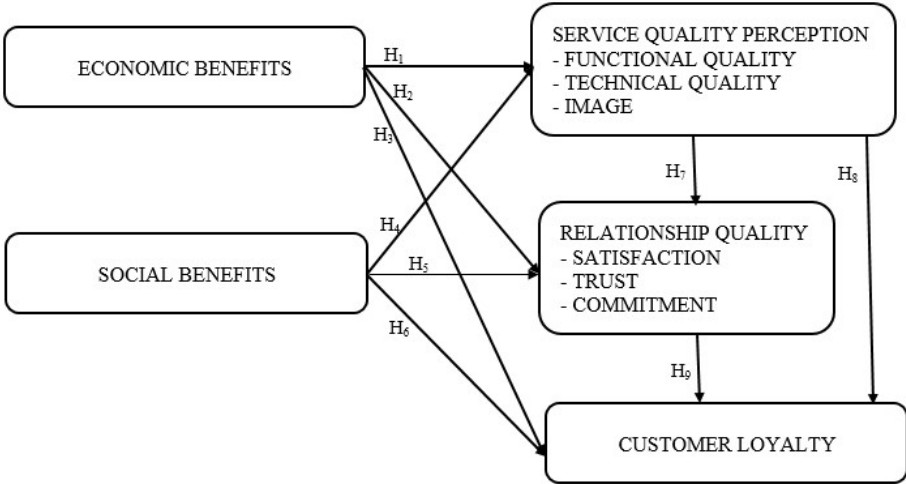

**Figure 1.** Hypothesized relationships of the model.

**4. Research Methodology**

The study uses mixed research methods, combining qualitative research methods and quantitative research methods. The study needs to synthesize and generalize relevant

theories to identify research concepts in the context of researching the university–business relationship in HCMC, forming a research model, and building a scale measure. Therefore, it is necessary to carry out some qualitative research to adjust and identify the variables of the model in order to build a theoretical framework for the problem to be studied and to build and calibrate the scales of research concepts. Quantitative research methods are used to confirm and re-check the research results of the qualitative research, identify variables and correlations between variables, and quantify the impact of variables in the research models. Quantitative research is employed on data from survey questionnaires and surveys. Main research steps:

- Preliminary research includes two studies: (1) qualitative research including in-depth interviews ($n = 6$) and two focus group ($n = 16$ and $n = 15$) for the purpose of model formation and building, and calibrating scales for concepts, (2) preliminary quantitative research is done through direct interview technique with sample size $n = 114$. Thereby, the scale is preliminarily assessed for reliability with Cronbach's Alpha.
- The main study was carried out using a quantitative method. The study has a total of 46 initial estimated parameters, the research program aims to collect over 460 questionnaires. Through the questionnaire survey, 486 valid answers were collected. The data are processed and analyzed using the software SPSS 20 and AMOS 24. The main research is used to confirm the reliability and validity of the scales and to test the research model. The scale is officially evaluated through reliability assessment Cronbach's Alpha and exploratory factor analysis (EFA) to evaluate the scale's value. The scale model is tested for validity and reliability through confirmatory factor analysis (CFA), and the theoretical model is tested through SEM linear structural modeling analysis method, tested with Bootstrap. In addition, the study also conducted a multi-group analysis

### 4.1. Selection of Respondents

The sample was selected according to the convenience sampling method with two control attributes: (1) Having a service relationship with at least one higher education institution, (2) The place of business is HCMC. The interviewees are leaders or managers of enterprises operating in Ho Chi Minh City who have a service relationship with the university; Each enterprise only interviewed one person.

As mentioned, by the end of 2020, Ho Chi Minh City had 254,699 operating enterprises. Data at the end of 2019 of the White Paper on the size of Vietnamese enterprises showed that only 2.6% are large enterprises and 3.4% are medium enterprises, the remaining 94% are small and micro enterprises. However, currently, there are no data on the percentage or list of enterprises in HCMC that have a relationship with the university. Due to the lack of data, the study had to collect samples in a convenient method based on the author's accessibility.

### 4.2. Collection of Responses and Sample Size

In the preliminary research, the author combined face-to-face interviews and mailing: The author participated in enterprise meetings at the university and asked for direct interviews. In addition, the author asked for information about businesses at these events to send letters. This method was done before the lockdown time due to COVID-19, 2020. The result was 114 valid answer sheets.

In the main survey, the author collected samples by convenience method. The author received support from organizations such as the Vietnam Chamber of Commerce and Industry—HCMC Branch (VCCI), HCMC Business Association (HUBA), and other Associations and Business Clubs in HCMC and the help of universities help send the survey in Google Form via email and social media groups of the above Organizations. The total number of subjects who are members of these groups was about 10,000 enterprises. Survey period: From April 2020 to July 2020. As a result, 486 valid answer sheets were obtained.

The sample size needs to be large enough to guarantee the necessary confidence estimation of the linear structural model (SEM) (Raykov and Widaman 1995) and needs to be considered in relation to the number of estimated parameters. (Hair et al. 2010). Bollen (1989) suggested that there should be a minimum of five observations per estimator (ratio 5:1). However, Kline (2005) suggests that this ratio should be 10:1. This study had 46 estimated parameters; so the ratio is 486: 46 > 10, reaching the threshold required by Kline (2005).

### 4.3. Sample Characteristic

The total number of survey questionnaires with complete responses were 486, of which 40.10% were in a relationship of 5 years or less and 59.90% had a relationship for more than 5 years. Divided by the equity of enterprises, 71% was private enterprises and 28% was state enterprises. Sorted from low to high by business's annual sales, 48% are under $1 million, 25% are between $1 and $4 million, and 27% are over $4 million.

### 4.4. Scale for Measuring Research Concepts

Referring to the scales of previous studies, the author carried out qualitative research to adjust the scales to suit the research context. There are five main concepts considered in the model, including (1) Perceived service quality (PSQ) (including three components: technical quality, functional quality, and image), (2) Economic benefits (ECB), (3) Social benefits (SOB), (4) Relationship quality (RQ) (including three components are satisfaction, trust, and commitment), and (5) Customer loyalty (LOY). All constructs were measured on a five-point scale, ranging from strongly disagree with statement (1) to strongly agree with statement (5). The scale is presented in Appendix A (Table A1).

## 5. Results

Through testing reliability by Cronbach's alpha, two observed variables SAT4 and TRU4 were excluded due to variable-total correlation coefficient < 0.30. EFA analysis results showed: With three independent variables perceived service quality (PSQ), economic benefits (ECB), and social benefits (SOB), the results show that the coefficient KMO = 0.916 > 0.5, KMO and Barlett's test in factor analysis results sig = 0.000, at eigenvalue = 1.674, the extracted variance was 62.449% (>50%) and five components are extracted. However, IMQ4 was rejected because it did not reach the discriminant value. After removing IMQ4 and re-analyzing, the results showed that KMO coefficient = 0.915 > 0.5, sig = 0.00, at eigenvalue = 1.656, extracted variance was 63.484% (>50%) extracted five components, in where the 3 components of the variable PSQ are segregated into three different groups. For the intermediate variable RQ, the results of EFA analysis showed that the coefficient KMO = 0.873 > 0.5, sig = 0.00, at eigenvalue = 1783, the extracted variance is 60.924% > 50%, and two components are extracted, in which the observed variables belonging to two groups SAT and TRU grouped into one were named SATTRU. For the dependent variable LOY, the results of EFA analysis showed that the coefficient KMO = 0.875 > 0.5, sig = 0.00, at eigenvalue = 3300 variance extracted 66.005% (>50%) extracted 1 component. Thus, the results of the EFA analysis showed that all indicators met the requirements. In the CFA analysis, after hooking e3 (SAT3)—e7 (TRU5) and removing SOB6, the critical model CFA results with 42 observed variables showed that the model was satisfactory with Chi squared = 1,106,558 with 790 degrees of freedom, $p = 0.000$, CMIN/df = 1.401 < 2, RMSEA = 0.029 < 0.05 and GFI = 0.903; CFI = 0.970; TLI = 0.967 > 0.9. The results of the convergence test show that all the weights of the variables were > 0.5 and were statistically significant at the 99.9% level (Anderson and Gerbing 1988). Check discriminant value, the results of correlation analysis showed that the correlation of the variables was < 1 and the difference was statistically significant.

The results of the analysis of the combined reliability coefficient and the extracted total variance showed that the combined confidence coefficient (CR) of the latent variables were

both higher than 50%, and the AVE indices were all larger than MSV. Thus, the fit indexes of the model were satisfactory.

### 5.1. Testing the Theoretical Model

Through testing the scales of this research model, the results were appropriate. The test results by the structural equation model showed that Chi-squared = 1,158,004 with 803 degrees of freedom; $p$ = 0.000; CMIN/df = 1.442 < 2; RMSEA = 0.030 < 0.05; GFI = 0.899 was close to 0.9; the indexes CFI = 0.966 and TLI = 0.963 were both > 0.9. Therefore, it can be concluded that the model fits the market data. Figure 2 presents the abbreviated SEM results.

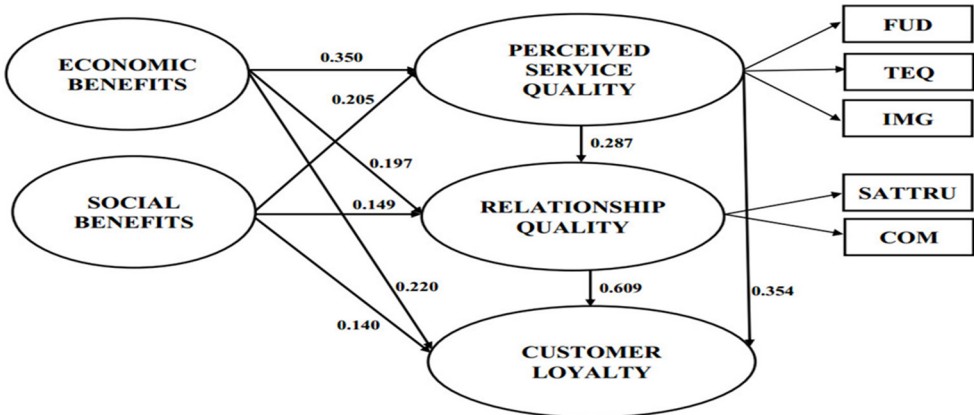

**Figure 2.** SEM results. Source: Official quantitative research results.

The results in Table 1 showed that the hypothetical relationships were statistically significant at the 5% level of significance.

**Table 1.** SEM test results.

|  |  |  | Estimate | S.E. | C.R. | *p* | Label |
|---|---|---|---|---|---|---|---|
| PSQ | <— | ECB | 0.350 | 0.061 | 5.718 | *** |  |
| PSQ | <— | SOB | 0.205 | 0.051 | 4.056 | *** |  |
| RQ | <— | PSQ | 0.287 | 0.082 | 3.485 | *** |  |
| RQ | <— | ECB | 0.197 | 0.050 | 3.984 | *** |  |
| RQ | <— | SOB | 0.149 | 0.039 | 3.842 | *** |  |
| LOY | <— | PSQ | 0.354 | 0.131 | 2.706 | 0.007 |  |
| LOY | <— | ECB | 0.220 | 0.073 | 3.006 | 0.003 |  |
| LOY | <— | SOB | 0.140 | 0.058 | 2.425 | 0.015 |  |
| LOY | <— | RQ | 0.609 | 0.219 | 2.778 | 0.005 |  |

Source: Official quantitative research results. Notes: *** $p < 0.001$.

Thus, the estimation results in Figure 2 and Table 1 showed that the hypothesized relationships in the theoretical model had a p-level of significance varying from 0.000 to 0.05, reaching the necessary level of significance (confidence interval 95%), all 9 hypotheses were accepted

### 5.2. Bootstrap Estimation

Estimation results by Bootstrap in Table 2 show that most of the deviations were very small and not statistically significant. Therefore, it can be concluded that the estimates in the model can be trusted.

**Table 2.** Estimation results by Bootstrap with $N = 1000$.

| Parameter | | | Estimate | SE | SE-SE | Mean | Bias | SE-Bias | CR | $p$ | Conclusion |
|---|---|---|---|---|---|---|---|---|---|---|---|
| PSQ | <— | ECB | 0.457 | 0.071 | 0.002 | 0.453 | −0.004 | 0.003 | −1.333 | 0.1830 | STABLE |
| PSQ | <— | SOB | 0.307 | 0.089 | 0.003 | 0.309 | 0.002 | 0.004 | 0.500 | 0.6173 | STABLE |
| RQ | <— | PSQ | 0.365 | 0.126 | 0.004 | 0.373 | 0.007 | 0.006 | 1.167 | 0.2439 | STABLE |
| RQ | <— | ECB | 0.328 | 0.091 | 0.003 | 0.324 | −0.004 | 0.004 | −1.000 | 0.3178 | STABLE |
| RQ | <— | SOB | 0.284 | 0.086 | 0.003 | 0.282 | −0.002 | 0.004 | −0.500 | 0.6173 | STABLE |
| LOY | <— | PSQ | 0.256 | 0.157 | 0.005 | 0.243 | −0.013 | 0.007 | −1.857 | 0.0639 | STABLE |
| LOY | <— | ECB | 0.209 | 0.087 | 0.003 | 0.202 | −0.007 | 0.004 | −1.750 | 0.0808 | STABLE |
| LOY | <— | SOB | 0.152 | 0.084 | 0.003 | 0.145 | −0.007 | 0.004 | −1.750 | 0.0808 | STABLE |
| LOY | <— | RQ | 0.346 | 0.221 | 0.007 | 0.365 | 0.019 | 0.010 | 1.900 | 0.0580 | STABLE |

Source: Official quantitative research results.

### 5.3. Multigroup Analysis

As mentioned, 94% of enterprises in Vietnam are small and micro enterprises, so the study did not select multi-group analysis by enterprise size. In addition, at present, enterprises in Vietnam operate under the market economy with business relationships that do not distinguish between state-owned and private enterprises, so the study does not choose multi-group analysis by corporate equity. Meanwhile, in Asian culture, the treatment of long-term partners is often different from the treatment of new partners. However, in the context of economic transformation in Vietnam, there may be changes to explore. In this study, the multi-group analysis method was used to compare the research model over time relationship. Relationship time is divided into two groups: (1) relationship time from 1–5 years and (2) long relationship time > 5 years. The multi-group analysis method in this study included the variable model and the partial invariant model. In the variable method, the estimated parameters in each model were not constrained, the relationship between the concepts in the model had different values between groups. In the partially invariant model, the measure component was not constrained, but the relationships between the concepts in the model were equally valid for the groups (see Table 3).

**Table 3.** Estimation of the variable model.

| | | | Regression Weights: (1–5 Years) | | | | Regression Weights: (>5 Years) | | | |
|---|---|---|---|---|---|---|---|---|---|---|
| | | | Estimate | S.E. | C.R. | $p$ | Estimate | S.E. | C.R. | $p$ |
| PSQ | <— | ECB | 0.341 | 0.06 | 5.679 | *** | 0.341 | 0.060 | 5.679 | *** |
| PSQ | <— | SOB | 0.202 | 0.05 | 4.164 | *** | 0.202 | 0.049 | 4.164 | *** |
| RQ | <— | PSQ | 0.324 | 0.08 | 4.088 | *** | 0.324 | 0.079 | 4.088 | *** |
| RQ | <— | ECB | 0.136 | 0.04 | 3.285 | 0.001 | 0.136 | 0.041 | 3.285 | 0.001 |
| RQ | <— | SOB | 0.141 | 0.03 | 4.231 | *** | 0.141 | 0.033 | 4.231 | *** |
| LOY | <— | PSQ | 0.538 | 0.12 | 4.414 | *** | 0.538 | 0.122 | 4.414 | *** |
| LOY | <— | ECB | 0.254 | 0.06 | 3.985 | *** | 0.254 | 0.064 | 3.985 | *** |
| LOY | <— | SOB | 0.200 | 0.05 | 3.963 | *** | 0.200 | 0.051 | 3.963 | *** |
| LOY | <— | RQ | 0.281 | 0.13 | 2.217 | 0.027 | 0.281 | 0.127 | 2.217 | 0.027 |

Source: Official quantitative research results. Notes: *** $p < 0.001$.

Comparing the difference in model compatibility index, the Chi-square test was used to compare the two models. The results of Table 4 show that $p = 0.0860 > 0.05$, so we chose the invariant model, the model with higher degrees of freedom. The study chose the

invariant model to read the results because it has higher degrees of freedom. Conclusion: there is no difference in the impact of variables in the model between respondents with different relationship lengths.

**Table 4.** Different invariant model compatibility criteria.

| Model | Chi-Square | df |
|---|---|---|
| Invariant | 2118.099 | 1615 |
| Mutable | 2102.914 | 1606 |
| Different | 15.185 | 9.000 |
| Conclude | 0.085978086 | Choose an invariant model |

Source: Official quantitative research results.

Besides the direct relationships shown in Figure 2 and Table 1, the research results also showed indirect relationships as shown in Table 5.

**Table 5.** Direct, indirect, and total impacts.

| | Impact | ECB | SOB | PSQ | RQ |
|---|---|---|---|---|---|
| PSQ | Direct | 0.350 | 0.205 | | |
| RQ | Direct | 0.197 | 0.149 | 0.287 | |
| | Indirect | 0.100 | 0.059 | | |
| | Total | 0.297 | 0.208 | 0.287 | |
| LOY | Direct | 0.220 | 0.140 | 0.354 | 0.609 |
| | Indirect | 0.195 | 0.127 | 0.175 | |
| | Total | 0.415 | 0.267 | 0.529 | 0.609 |

Source: Official quantitative research results.

## 6. Discussing Research Results

Research results showed that the research model fits the market data, all nine proposed hypotheses are accepted. In which, economic benefits (ECB) and social benefits (SOB) have a significant direct influence on perceived service quality (PSQ), and have both direct and indirect effects on the quality of university–enterprise relationship (RQ) and loyalty of enterprises (LOY). PSQ has a significant direct influence on the RQ, and at the same time has a direct and indirect influence on LOY, and RQ has the greatest direct influence on LOY.

Regarding the degree of influence of the relational benefits on the remaining factors of the model, although previous studies on the benefits of B2B relationships have confirmed that social benefits always have a high degree of influence and significantly stronger effects than economic benefits (with economic benefits being included in benefits receiving special treatment according to the classification applied in Gwinner et al. (1998) B2C relationship studies) (e.g., Palaima and Auruškevičienė 2007; Gremler et al. 2020); this study confirms that the opposite has happened in the university–enterprise relationship in HCMC, Vietnam.

- The degree of the direct influence of the ECB on the PSQ of 0.350 is larger than the effect of the social benefits on the PSQ of 0.205.
- The degree of the direct influence of the ECB on the RQ of 0.197 is larger than the direct influence of the SOB on the RQ of 0.149.
- The degree of ECB's Indirect Effect on RQ of 0.100 is larger than SOB's Indirect Effect on RQ of 0.059.
- The degree of the total influence of the ECB on RQ of 0.297 is greater than the sum of the influence of the SOB on the RQ of 0.208.
- The degree of the direct influence of the ECB on the LOY of 0.220 is larger than the direct influence of the SOB on the LOY of 0.140.

- The degree of ECB's Indirect Effect on LOY of 0.195 is larger than that of SOB's Indirect Effect on LOY of 0.127.
- The degree of the ECB's total influence on LOY of 0.415 is significantly larger than the total effect of SOB on LOY of 0.267.

Thus, the quantitative research results have confirmed the qualitative research results that for businesses that have a "relationship" with universities in Ho Chi Minh City, Vietnam, compared to social benefits, economic benefits have a significantly stronger influence on relationship quality and customer loyalty. This result supports Peterson (1995); Sheth and Parvatiyar (1995) and Gwinner et al. (1998), these scholars both argue that "the economic benefits that customers can derive from the relationship are the main drivers for developing B2B relationships". This is a new finding in empirical research on the benefits of B2B relationships in the service sector.

Besides, the acceptance of hypotheses H1 and H4 "economic benefits and social benefits have a significant direct impact on perceived service quality" has supported the qualitative research results of the author, confirming that when enterprises receive economic and social benefits, they have a more positive perception of the university's service quality. This is also a new finding of the study. This result is consistent with the "feelings shape thought and thought shapes feelings" views of Isen and Baron (1991), as well as supporting the views of George and Brief (1996) and Podsakoff and MacKenzie (1997), who argue that feelings and emotions influence several important organizational behaviors.

In addition, the research results also showed that, besides directly affecting the quality of university–enterprise relationship (RQ) and business loyalty (LOY), perceived service quality (PSQ) is also an intermediate variable that increases the influence of economic benefits (ECB) and social benefits (SOB) on RQ and LOY. Similarly, besides having a direct effect on LOY, RQ also acts as an intermediate variable that significantly increases the influence of ECB, SOB, and PSQ on LOY.

There are two unexpected results of the study. Firstly, in theory, the components of the university–enterprise relationship quality concept include three components: satisfaction, trust, and commitment. However, in this study, the two components satisfaction (SAT) and trust (TRU) have high intrinsic unity and converge into one component SATTRU. This convergence may be due to the way it is represented in the research context. In fact, trust is very important in business relations in Vietnam, enterprises only trust universities that they are really satisfied with after experiencing previous transactions. When satisfied and confident with the university, enterprises often perceive this university as having a good reputation instead of expressing "I am satisfied and confident in this university". This represents a different perception of the concept of RQ in the context of research on B2B relationship quality in a particular service sector in HCMC, Vietnam. Secondly, the results of the multi-group analysis showed that the effects of the model did not differ over the relationship time. This result implies that for enterprises, the length of the relationship with the university does not change the interest in receiving benefits towards pragmatism; Compared with previous Asian cultural trading practices, this result could be one of the changes in an economy in transition.

## 7. Conclusions and Implications

With the aim of comparing the influence of economic benefits with social benefits in the model of integrating customer benefits and relationship quality in the context of university–enterprise relationship research in HCMC, Vietnam, from the perspective of enterprises, a study combining qualitative and quantitative research was carried out. The research results indicated that economic benefits and social benefits have a direct influence on the perceived service quality, relationship quality, and customer loyalty. The results of examining the influence of each of these types of relational benefits in the research model showed that the influence of economic benefits is significantly stronger than the influence of social benefits. This is a new finding of this study compared to previous studies on relational benefits. Besides that, the direct influence of economic benefits and social benefits on perceived service quality is also a new finding.

The research results mentioned above imply that universities that bring better economic benefits and social benefits to enterprises will be perceived more positively by enterprises in terms of service quality and relationship quality with enterprises will be better and businesses will be more loyal. It should be noted that "economic benefits" have a significantly stronger influence on other factors than "social benefits". Besides, the university in which service quality is better perceived by enterprises, the quality of the relationship with enterprises will be better and enterprises will be more loyal. In addition, perceived service quality is also an intermediate variable that increases the influence of economic and social benefits on relationship quality and loyalty of enterprises. The research results also imply that in order to gain the loyalty of enterprises, universities need to pay special attention to improving the quality of relationships with enterprises. Finally, the university should note that the length of the relationship does not change the interest in receiving benefits according to the pragmatic orientation of enterprises.

The study contributed to enhancing the understanding of the important role of the benefits of enterprises in the university–enterprise relationship by determining the degree of influence of these benefits on the quality of the relationship and corporate loyalty. The test results showed that the theoretical model fits the market data and the research hypotheses are accepted. This result can have practical implications for various audiences such as higher education institutions in Vietnam, service providers, enterprises using university services, and academic researcher in the field of relationship marketing.

## 8. Limitations and Future Research

This study has several limitations. First, the study only examines the effects of ECB and SOB on PSQ, RQ, and LOY in the context of a B2B relationship in a new service sector; Although this study confirmed the importance of concepts related to theoretical modeling, there may be other statistically significant relational benefits that need to be explored. Besides, this study was only conducted in HCMC, the generalizability of the research results would be higher if it was repeated with the sample structure including enterprises in other big cities of Vietnam. In addition, although the results showed that the model has a high fit; however, the generalizability of the research model will be higher if it is repeated in the study of B2B relationships in another service sector or another industry. Finally, this study was conducted from the perspective of enterprises, in order to better understand the relationship between universities and enterprises, more research may be needed to approach this issue from the perspective of universities.

Future studies may consider extending and testing this model with other relational benefits. Besides, further studies can repeat the model with a sample structure including enterprises in other big cities of Vietnam or repeat this research model in other service sectors or other industries. In addition, future studies can approach the issue of customer benefits and relationship quality from the perspective of the university.

**Author Contributions:** Conceptualization, P.T.N., H.M.P. and H.V.C.; methodology, P.T.N., H.M.P. and H.V.C.; software, P.T.T.; validation, P.T.N., H.M.P. and H.V.C.; formal analysis, P.T.N. and P.T.T.; investigation, P.T.N. and P.T.T.; resources, H.V.C.; data curation, P.T.T.; writing—original draft preparation, P.T.N.; writing—review and editing, P.T.N.; visualization, P.T.T.; supervision, H.M.P.; project administration, H.V.C. All authors have read and agreed to the published version of the manuscript.

**Funding:** This work was supported by Grant No. 2022.01.28 from Nguyen Tat Thanh University, Ho Chi Minh City, Vietnam.

**Institutional Review Board Statement:** Not applicable.

**Informed Consent Statement:** Not applicable.

**Data Availability Statement:** No new data were created or analyzed in this study. Data sharing is not applicable to this article.

**Conflicts of Interest:** The authors declare no conflict of interest.

## List of Abbreviations

| Acronyms | Acronym Meaning |
|---|---|
| COM | Commitment |
| ECB | Economic Benefits |
| FUQ | Functional Quality |
| IMQ | Image |
| LOY | Customer Loyalty |
| PSQ | Perceived Service Quality |
| RB | Relational Benefits |
| RM | Relationship Marketing |
| RQ | Relationships Quality |
| SAT | Satisfaction |
| SOB | Social Benefits |
| TEQ | Technical Quality |
| TRU | Trust |

## Appendix A

**Table A1.** Scale of Research Concepts.

| Code | Dimension | Questionnaire Statement | Source |
|---|---|---|---|
| ECB1 | Economic Benefits | "XYZ university offers a discounted price/service fee for my company thanks to its relationship with XYZ university". | Nguyen et al. (2021) adapted from Williamson (1988); Heide and John (1992); Kalwani and Narayandas (1995); Sheth and Parvatiyar (1995); Gwinner et al. (1998); and Li et al. (2012). |
| ECB2 | Economic Benefits | "My company saves time in searching for other service providers thanks to its relationship with XYZ university". | Nguyen et al. (2021) adapted from Gwinner et al. (1998). |
| ECB3 | Economic Benefits | "Thanks to the relationship with XYZ university, my company is able to recover R & D costs faster than without the relationship". | Nguyen et al. (2021) adapted from Sheth and Parvatiyar (1995). |
| ECB4 | Economic Benefits | "XYZ university is willing to share technology, information and market access opportunities with my company thanks to its relationship with XYZ university". | Nguyen et al. (2021) adapted from Wilson (1995). |
| ECB5 | Economic Benefits | "Thanks to the relationship, my Company can get useful knowledge updates from XYZ university". | Nguyen et al. (2021) adapted from Badaracco (1991); and Wilson (1995). |
| ECB6 | Economic Benefits | "Thanks to the relationship, my Company can receive special additional services of XYZ university" | Nguyen et al. (2021) adapted from Gwinner et al. (1998); and Gremler et al. (2020). |
| SOB1 | Social Benefits | "Leader of my company is invited to attend and honor in XYZ university events". | |
| SOB2 | Social Benefits | "Leader of my company like certain social aspects of the relationship with XYZ university (enjoy participating in the educational environment, showing corporate social responsibility, . . . )" | |
| SOB3 | Social Benefits | "Leader of my company have developed a friendship with XYZ university's representatives". | Nguyen et al. (2021) adapted from Gwinner et al. (1998); and Hennig-Thurau et al. (2002). |
| SOB4 | Social Benefits | "Leaders and representatives of XYZ university know the name of the leaders of my company". | |
| SOB5 | Social Benefits | "Through the relationship with XYZ university, my company is able to access other business opportunities". | |
| SOB6 | Social Benefits | "The relationship with XYZ university helps to increase my company's brand awareness". | |

**Table A1.** *Cont.*

| Code | Dimension | Questionnaire Statement | Source |
|------|-----------|------------------------|--------|
| TEQ1 | Technical quality | The capacity of XYZ university graduates meets the requirements of my company. | Phuong et al. (2022) adapted from Grönroos (1993, 2000). |
| TEQ2 | Technical quality | XYZ university's training courses exclusively developed for my company help the company improve the quality of human resources. | |
| TEQ3 | Technical quality | The applied/technological studies transferred by XYZ university are useful to my company. | |
| TEQ4 | Technical quality | XYZ university has a strong and secure information technology system, which helps to quickly and smoothly fulfill my company's orders. | |
| TEQ5 | Technical quality | XYZ university applies technological advancements to provide useful technical solutions for my company. | |
| FUQ1 | Functional quality | XYZ university shows an interest in my company's interests. | Phuong et al. (2022) adapted from Sharma and Patterson (1999); Palaima and Auruškevičienė (2007); and Auruškevičienė et al. (2010). |
| FUQ2 | Functional quality | XYZ university leaders cherish the relationship with my company. | |
| FUQ3 | Functional quality | XYZ university is very accessible when my company needs to provide services. | |
| FUQ4 | Functional quality | When providing services, XYZ university seeks to communicate with my company's employees | |
| FUQ5 | Functional quality | XYZ university's representatives respond promptly to my company's requests / questions. | |
| FUQ6 | Functional quality | I highly appreciate the hospitality of XYZ university's representatives and staff. | |
| IMQ1 | Image | XYZ university has a good reputation. | Phuong et al. (2022) adapted from Grönroos (1993, 2000). |
| IMQ2 | Image | XYZ university is sincere with my company | |
| IMQ3 | Image | I have a good experience using XYZ university's services. | |
| IMQ4 | Image | XYZ university has great contributions to the society | |
| SAT1 | Satisfaction | "We are satisfied with the services provided by XYZ university". | Nguyen et al. (2021) adapted from Crosby et al. (1990); Ling and Ding (2006); and Liu et al. (2011) |
| SAT2 | Satisfaction | "We are completely satisfied with the processes and procedures that XYZ university has done with us". | |
| SAT3 | Satisfaction | "The communications between my company and the representative of XYZ university always make us feel satisfied". | |
| SAT4 | Satisfaction | "Overall, I think XYZ university is a good service provider". | |
| TRU1 | Trust | "The staff of XYZ university follow what XYZ university promises to my company". | Nguyen et al. (2021) adapted from Crosby et al. (1990); Morgan and Hunt (1994); Ulaga and Eggert (2004); Wong and Sohal (2006); and Auruškevičienė et al. (2010) |
| TRU2 | Trust | "I believe that XYZ university considers the best benefit of my company". | |
| TRU3 | Trust | "I feel that I can always trust XYZ university". | |
| TRU4 | Trust | "I believe XYZ university will do everything correctly". | |
| TRU5 | Trust | "XYZ university's staff are honest". | |

**Table A1.** *Cont.*

| Code | Dimension | Questionnaire Statement | Source |
|------|-----------|------------------------|--------|
| COM1 | Commitment | "The relationship with XYZ university is very important to our operations". | Nguyen et al. (2021) adapted from Morgan and Hunt (1994); Hennig-Thurau et al. (2002); Wong and Sohal (2006); Ulaga and Eggert (2004); and Auruškevičienė et al. (2010) |
| COM2 | Commitment | "The relationship with XYZ university is worthy of my company's highest effort to maintain". | |
| COM3 | Commitment | "We will maintain the current relationship with XYZ university for an infinite time". | |
| COM4 | Commitment | "Our relationship with XYZ university is like a family". | |
| LOY1 | Customer Loyalty | "My company hopes to expand the scope of cooperation with XYZ university". | Nguyen et al. (2021) adapted from Sharma and Patterson (1999); and Palaima and Auruškevičienė (2007). |
| LOY2 | Customer Loyalty | "My company intends to develop more projects with XYZ university". | |
| LOY3 | Customer Loyalty | "Most likely in the nearest future, we will choose another partner university". (reversed) | |
| LOY4 | Customer Loyalty | "I would recommend XYZ university to other companies". | |
| LOY5 | Customer Loyalty | "If someone tells me that the quality of XYZ university's provided service is poor, I will try to prove that it is not true". | |

All constructs are measured on a five-point scale, ranging from strongly disagree with statement (1) to strongly agree with statement (5).

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
