# Peer review of "Effects of Relational Benefits in the Model of Customers’ Benefits and Relationship Quality in Vietnam"

_economies, doi:10.3390/economies10110283_

Round 1

Reviewer 1 Report

Abstract need more details of the whole study

Introduction

Various studies have been conducted on these concepts, what is new in this study?

What sector or industry or any specific products or services that has been undertaken in this study

Explain the gaps and justify the objectives of the study

Explain the core issues and the evidence especially statistics and with current literatures to support the issues of this study

Any recent statistics from the officials to support the issue or arguments

Identify the research problem -- as with any academic study, you must state clearly and concisely the research problem that is being investigated.

Appropriate background information has not been provided with special terms and concepts defined.

Lacks research topic or problem not clearly stated shown to be worth investigating as there were many studies conducted, and therefore need to highlight the extension of this study from previous studies

Introduce the topic, explain its relevance to the audience, state a thesis or purpose, and outline the main points.

Citations in wrong format

Literature Review

Need to strengthen the literature review

Places each source in the context of its contribution to the understanding of the specific issue, area of research, or theory under review.

Describes the relationship of each source to the others that you have selected (IVs & DV)

Identifies new ways to interpret, and shed light on any gaps in, previous research

Review scholarship on the topic, synthesizing key themes, and, if necessary, noting studies that have used similar methods of inquiry and analysis.

Note where key gaps exist and how your study helps to fill these gaps or clarifies existing knowledge.

The author should have discussed the issues in detail and how the issues were not resolved or partially resolved by previous studies.

Discuss the important recent extensions that have been made to the model to make it more realistic, and give a brief overview of some of the older and more recent empirical studies that have fitted the model. Such contributions add up the value of the paper

Independent variables and dependent variables were not discussed through and the relationship. Any relationship between the variables and theories also need to be discussed

Literature review is lacking the in-depth of the study

Separate the paragraphs according o the variables of the study and provide the detail explanation with the gaps

Extensive explanation of IVs & DV is required to justify the importance of the variables of the study and to sync with the items in questionnaire. Some of the questions does not match the explanation in literature review

Explain all the variables thoroughly in literature review and how these variables are related to the study and able to resolve the problems and achieve the objectives of the study

No theory to support the variables of study, what is the underpinning theory that explains the logic of the variables, The variables need to be explained in detail in literature review and explain the relationship and relate it to the relevant theories that supports the study

Methodology

State clearly in research method on how data has been extracted through surveys and interviews

Only in research methodology SMEs has been introduced, is this study about SMEs  and if it is about SMES, why haven’t it has been introduced in introduction and literature review

What is HCMC? Too many jargons used and unable to justify those jargons, suggest to have a tables for all jargons in appendix

Explain and justify why survey and interviews needed, Is this going to be triangulation research

emails was sent to about 10,000 businesses, how long did it take to complete the survey and what was the response like

What is the rationale of sample size

Is the population known or unknown for sampling strategies  

If population known, state the number and do the samples calculation

What scales has been utilised such as ordinal, nominal interval etc

Why pilot test has not been conducted to analyse reliability of the variables and was the questionnaire adopted or adapted

Lack of evidence of care and accuracy in the data process

Unable to reveal the research methods fully described of the advantages and disadvantages of chosen methods that was discussed.

Analysis

The organization and discussion could be improved quite a bit, to make it clearer in some places to demonstrate symbolic role

Important – need to append the SEM software model as appendix to substantiate all the explanation in analysis part

Lack of detailed analysis which does not answer the objectives which was not clearly mentioned

Need more explanation of the analysis to support the hypothesis

Discussion

Discussion has not been incorporated in the study

The finding of the research needs to be compared and contrasted with findings, theories, models and concepts derived from the literature review.

The relevance of the conclusions for stakeholders has not been discussed thoroughly

Comment on whether or not the results were expected for each set of findings; go into greater depth to explain findings that were unexpected or especially profound. If appropriate, note any unusual or unanticipated patterns or trends that emerged from your results and explain their meaning in relation to the research problem.

Either compare your results with the findings from other studies or use the studies to support a claim. This can include re-visiting key sources already cited in your literature review section

Describe the patterns, principles, and relationships shown by each major findings and place them in proper perspective. The sequence of this information is important; first state the answer, then the relevant results, then cite the work of others. If appropriate, refer the reader to a figure or table to help enhance the interpretation of the data

Good discussion section includes analysis of any unexpected findings. This part of the discussion should begin with a description of the unanticipated finding, followed by a brief interpretation as to why you believe it appeared and, if necessary, its possible significance in relation to the overall study.

The discussion section should end with a concise summary of the principal implications of the findings regardless of their significance. Give a brief explanation about why you believe the findings and conclusions of your study are important and how they support broader knowledge or understanding of the research problem.

Conclusion

The main research question has not been concisely answered.

The overall argument has not been summarized.

The reflection on the aims, methods, and results of the research is lacking

Relevant recommendations have not been discussed.

What are your recommendations to the stakeholders

The contributions of the research have not been clearly explained.

Conclusions and recommendations discussed in the context which are not widely applicable

Author Response

Dear Reviewer 1

I would like to sincerely thank all of your questions, comments, and suggestions. These questions and criticisms have shown your kindness; thanks to which I have tried to perfect my article.

However, due to limited capacity and limited time, even though I tried my best, I still could not answer each question as required, but integrated all the answers through the edited article. I enclose here the edited article and documents on quantitative processing.

We hope you will reconsider and advise further.

Once again, I would like to thank Mr/Ms.

Authors

Reviewer 2 Report

1. The novelty of the research needs to be presented specifically by explaining the contribution of this research to the role of economic benefit versus social benefit in accordance with the object of research.

2. The unique relevance of companies in Vietnam that have a relationship with higher education institutions needs to be explained in more detail to support their relevance to the research model built.

3. From the characteristics of the sample, the dominance in private enterprises (71% ) while for state-owned enterprises it is only 28%. Thus it is necessary to convey what are the consequences of the majority of these private companies in providing arguments against the research results found. Given the existence of different treatment for both types of enterprises in the economic model and the social model. Describe according to the conditions in the country of Vietnam (HCMH).

4. Testing for Multigroup analysis that divides based on the duration of the relationship, it is necessary to give arguments, why it is important and what are the consequences for research variables. Is it possible to see from the perspective of another multigroup that is very important, for example from the characteristics of the company (private vs enterprise) or the characteristics of the scale of the business. 

5. The explanation for B2B seems very short and there is no argument that prefaces the previous explanation. Does this only apply to B2B or can it be accelerated for B2C or marketplaces? This needs to be sharply conveyed.

Author Response

Dear Reviewer 2

I would like to sincerely thank all of your questions, comments, and suggestions. These questions and criticisms have shown your kindness; thanks to which I have tried to perfect my article.

However, due to limited capacity and limited time, even though I tried my best, I still could not answer each question as required, but integrated all the answers through the edited article. I enclose here the edited article and documents on quantitative processing.

We hope you will reconsider and advise further.

Once again, I would like to thank Mr/Ms.

Authors
